# Add-on effect of Chinese herbal medicine in the treatment of mild to moderate COVID-19: A systematic review and meta-analysis

Xuqin Du[1,2], Lipeng Shi [1,2]*, Wenfu Cao[1,2,3]*, Biao Zuo[1,2], Aimin Zhou[4]

**1** College of Traditional Chinese Medicine, Chongqing Medical University, Chongqing, People's Republic of China, **2** Chongqing Key Laboratory of Traditional Chinese Medicine for Prevention and Cure of Metabolic Diseases, Chongqing, People's Republic of China, **3** Department of Chinese Traditional Medicine, The First Affiliated Hospital of Chongqing Medical University, Chongqing, People's Republic of China, **4** Department of Cardiovascular U nit, Traditional Chinese medicine hospital Dianjiang Chongqing, Chongqing, People's Republic of China

* shilipeng0206@163.com (LS); caowenfu9316@163.com (WC)

## Abstract

### Introduction

Coronavirus disease 2019 (COVID-19) has emerged as a global pandemic since its outbreak in Wuhan, China. It is an urgent task to prevent and treat COVID-19 effectively early. In China's experience combating the COVID-19 pandemic, Chinese herbal medicine (CHM) has played an indispensable role. A large number of epidemiological investigations have shown that mild to moderate COVID-19 accounts for the largest proportion of cases. It is of great importance to treat such COVID-19 cases, which can help control epidemic progression. Many trials have shown that CHM combined with conventional therapy in the treatment of mild to moderate COVID-19 was superior to conventional therapy alone. This review was designed to evaluate the add-on effect of CHM in the treatment of mild to moderate COVID-19.

### Methods

Eight electronic databases including PubMed, EMBASE, Cochrane Central Register of Controlled Trials, the Clinical Trials.gov website, China National Knowledge Infrastructure (CNKI), China Science and Technology Journal Database (VIP), Wanfang Database and China Biology Medicine (CBM) were searched from December 2019 to March 2021 without language restrictions. Two reviewers searched and selected studies, and extracted data according to inclusion and exclusion criteria independently. Cochrane Risk of Bias (ROB) tool was used to assess the methodological quality of the included RCTs. Review Manager 5.3.0 software was used for statistical analysis.

### Results

Twelve eligible RCTs including 1393 participants were included in this meta-analysis. Our meta-analyses found that lung CT parameters [RR = 1.26, 95% CI (1.15, 1.38), *P*<0.00001]

**Data Availability Statement:** All relevant data are within the paper and its Supporting information files.

**Funding:** This work was supported by the National Natural Science Foundation of China

(No.81573860), Chongqing Medical University Postdoctoral Foundation (No. R11004), and Chongqing Postdoctoral Special Foundation (Yuren Social Office [2020] No. 379). The funders had no role in study design, data collection and analysis, decision to publish, or preparation of the manuscript.

**Competing interests:** The authors declare that they have no competing interests.

and the clinical cure rate [RR = 1.26, 95%CI (1.16, 1.38), $P<0.00001$] of CHM combined with conventional therapy in the treatment of mild to moderate COVID-19 were better than those of conventional therapy. The rate of conversion to severe cases [RR = 0.48, 95%CI (0.32, 0.73), $P = 0.0005$], TCM symptom score of fever [MD = -0.62, 95%CI (-0.79, -0.45), $P<0.00001$], cough cases [RR = 1.43, 95%CI (1.16, 1.75), $P = 0.0006$], TCM symptom score of cough[MD = -1.07, 95%CI (-1.29, -0.85), $P<0.00001$], TCM symptom score of fatigue[MD = -0.66, 95%CI (-1.05, -0.28), $P = 0.0007$], and CRP[MD = -5.46, 95%CI (-8.19, -2.72), $P<0.0001$] of combination therapy was significantly lower than that of conventional therapy. The WBC count was significantly higher than that of conventional therapy[MD = 0.38, 95%CI (0.31, 0.44), $P<0.00001$]. Our meta-analysis results were robust through sensitivity analysis.

## Conclusion

Chinese herbal medicine combined with conventional therapy may be effective and safe in the treatment of mild to moderate COVID-19. More high-quality RCTs are needed in the future.

## Introduction

Coronavirus disease 2019 (COVID-19) caused by severe acute respiratory syndrome coronavirus 2 (SARS-CoV-2) has emerged as a global pandemic since its outbreak in Wuhan, China, in December 2019 [1]. As of March 25, 2021, more than 124.21 million confirmed cases and more than 2.73 million deaths had been reported globally [2]. Unfortunately, confirmed cases continue to rise due to rapid spread. Thus, it is an urgent task to prevent and treat COVID-19 effectively early.

To date, the pandemic in China has been gradually controlled due to strong government measures, early detection, early quarantine, and early treatment with conventional Western therapy and Chinese herbal medicine (CHM) [3, 4]. CHM is a special medicine used in the prevention and treatment of diseases and is composed of plant medicine, animal medicine, and mineral medicine [5]. In China's experience combating the COVID-19 pandemic, CHM has played an indispensable role, and a CHM therapeutic schedule was included in the guidelines on the treatment of COVID-19 [4, 6]. A large number of epidemiological investigations have shown that mild to moderate COVID-19 accounts for the largest proportion of cases [7]. It is of great importance to treat such COVID-19 cases, which can help control epidemic progression. The current conventional therapy recommendations for mild to moderate COVID-19 are mainly antiviral and symptomatic support treatment [6]. The recommended antiviral drugs are interferon, ribavirin, lopinavir-ritonavir, and chloroquine phosphate [6]. However, most of the recommended antiviral drugs used to treat mild to moderate COVID-19 are currently based on previous treatments for severe acute respiratory syndrome (SARS) and influenza A, and uncertainties regarding the efficacy and side effects of these antiviral drugs remain problematic [8]. Many trials have shown that CHM combined with conventional therapy in the treatment of mild to moderate COVID-19 was superior to conventional therapy alone in improving clinical efficacy, clinical symptoms, and anti-inflammatory effects while causing fewer adverse drug events [9, 10].

Presently, there is no systematic evaluation report on the efficacy of CHM combined with conventional therapy in the treatment of mild to moderate COVID-19. Therefore, we performed a systematic review and meta-analysis of trials that tested the add-on effect of CHM in the treatment of mild to moderate COVID-19.

## Methods

The protocol for our review has been registered on the International Prospective Register of Systematic Reviews (PROSPERO) with the registration number CRD42020213528. This review was reported according to the Preferred Reporting Items for Systematic Reviews and Meta-Analyses (PRISMA) [11].

### Eligibility criteria

**Inclusion and exclusion criteria.** The diagnostic criteria of mild to moderate COVID-19 refer to the " Diagnosis and Treatment Guideline for COVID-19 (Trial 8th Edition) " [6]. Mild COVID-19 is defined as mild clinical symptoms (such as low fever, mild fatigue, impairment of smell and taste, etc.) with no radiographic evidence of pneumonia [6]. Moderate COVID-19 is defined as having fever, respiratory symptoms, and imaging manifestations of pneumonia [6].

Inclusion criteria: (1) Study design: only randomized controlled trials (RCTs). (2) Participants: adult patients (aged≥18 years) with an established diagnosis of mild to moderate COVID-19. (3) Interventions: the treatment group was treated with a combination of CHM and conventional therapy. The administration of CHM was limited to oral administration. Patients in the control group were treated with conventional therapy. (4) Outcomes: a. clinical efficacy (e.g. lung computed tomography (CT), clinical cure rate, rate of conversion to severe cases, viral nucleic acid testing), b. clinical symptoms (e.g. fever, cough, fatigue), c. inflammatory biomarkers (e.g. white blood cell (WBC) count, lymphocyte (LYM) count, C-reactive protein (CRP)), d. adverse drug events (e.g. nausea and vomit, diarrhea, liver damage).

Exclusion criteria: (1) Patients with suspected diagnosis of COVID-19; (2) Retrospective studies, observational studies, repeated data studies, and cross-over studies.

### Search strategy

Eight electronic databases including PubMed, EMBASE, Cochrane Central Register of Controlled Trials, the Clinical Trials.gov website, China National Knowledge Infrastructure (CNKI), China Science and Technology Journal Database (VIP), Wanfang Database and China Biology Medicine (CBM) were searched from December 2019 to March 2021 without language restrictions. The search terms included "coronavirus disease 2019", "COVID-19", "novel coronavirus pneumonia", "SARS-CoV-2", "2019-nCoV", "traditional Chinese medicine", "Chinese herbal medicine", "Chinese herb", "Chinese herb therapy", "Chinese herbal formulas", "clinical trial", "randomized controlled trial", "randomised controlled trial", and "lin chuang yan jiu". Potential eligible trials were obtained by searching the reference lists of reviews and meta-analyses. We also contacted with study authors for more information.

The PubMed search strategy is as follows. Search: ((((((coronavirus disease 2019) OR (COVID-19)) OR (novel coronavirus pneumonia)) OR (SARS-CoV-2)) OR (2019-nCoV)) AND ((((((traditional Chinese medicine) OR (Chinese herbal medicine)) OR (Chinese herb)) OR (Chinese herb therapy)) OR (Chinese herbal formulas))) AND ((((clinical trial) OR (randomized controlled trial)) OR (randomised controlled trial)) OR (lin chuang yan jiu)).

### Study selection and data extraction

Two reviewers (XQD and LPS) read the title, abstract, and full text, and selected the qualified trials according to the eligibility criteria independently. A pre-designed test form in duplicate was used for extracting the following information: basic characteristics (e.g. the title, first authors' name, publication date), participant characteristics (e.g. age, gender, sample size), intervention details (e.g. description of interventions, description of controls, dose, route of oral administration, duration of treatment), and outcome measures, as well as any adverse events. Reviewers (XQD and LPS) cross-checked the data. Any differences of opinion among the primary reviewers were resolved by a third reviewer (WFC). All reviewers were unbiased and had no conflicting interests.

### Assessment of methodological quality

Two reviewers (XQD and LPS) assessed the methodological quality by using the Cochrane Collaboration's tool [12]. Seven items of risk of bias (ROB) were evaluated as below: random sequence generation, allocation concealment, blinding (patient, investigator and assessor), incomplete outcome data addressed, free of selective reporting, and other biases. Each item of ROB was assessed to be low ROB, high ROB, or unclear ROB. Additionally, any disagreements of ROB were resolved by consultation with the third reviewer (WFC).

### Meta-analyses

Review Manager 5.3.0 software (The Cochrane Collaboration, Copenhagen, Denmark) was used for quantitative analysis. The relative risk (RR) was adopted for dichotomous variables. Mean difference (MD) or standard mean difference (SMD) were calculated for continuous variables. Confidence intervals (CIs) were set as 95% with $P < 0.05$ considered as a statistically significant difference. Heterogeneity was assessed with the $\chi 2$ test and the $I^2$ statistical value. When the $P \geq 0.10$ or $I^2 \leq 50\%$, a fixed-effect model was adopted. Otherwise, a random-effect model was applied. Subgroup analyses were carried out according to treatment duration. Sensitivity analyses were performed by leave-one-out method [13]. Funnel plot analysis was conducted to evaluate the reporting bias for outcome measures with more than 10 RCTs [14].

## Results

### Eligible studies

The flow diagram of study selection and identification is showed in (Fig 1). A total of 526 related citations were initially retrieved. Twelve eligible RCTs were included in meta-analysis according to the inclusion and exclusion criteria [15–26]. One RCT was published online in English [19], and the rest were reported online in Chinese.

The characteristics of included RCTs are listed in (Table 1). Twelve RCTs enrolling 1393 participants were included in this meta-analysis. All twelve RCTs were conducted in China in 2020. In all the studies included, the patients in control group received conventional therapy while patients in treatment group received combination therapy of CHM and conventional therapy. The treatment duration varied from 5 to 15 days. Among the twelve RCTs [15–26], three were multi-centered trials [18, 19, 22] and the remaining nine were single-centered trials.

### Assessment of methodological quality

The results of risk of bias assessment are shown in (Fig 2a) and (Fig 2b). In general, the quality of methodology included in this review was not high. Most of the RCTs did not clearly state

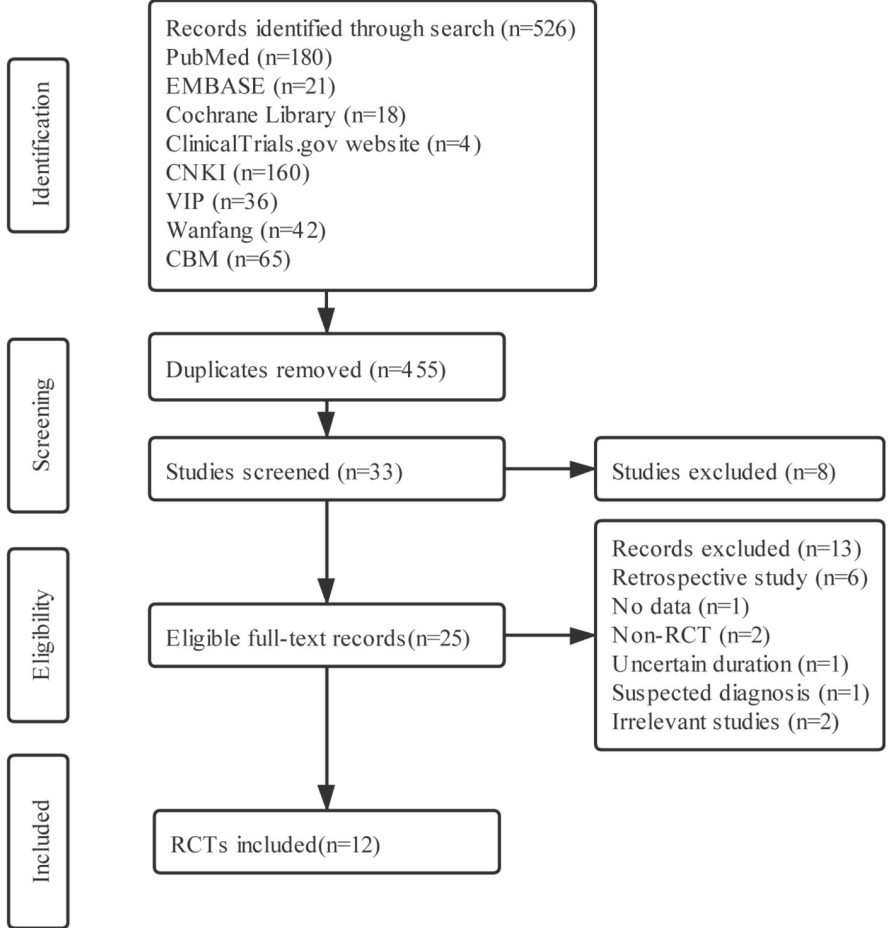

**Fig 1. The flow diagram of study selection and identification.**

detection bias, and all of them did not explicitly report allocation concealment, performance bias, and reporting bias.

## Description of CHM

The components of CHM are listed in (Table 2). Nine oral CHM were used in this review, including Jinhua Qinggan granule [15], Toujie Quwen granule [16, 17], Jinyinhua oral liquid [18, 25], Lianhua Qingwen capsule (granule) [19, 23], Maxing Xuanfei Jiedu Decoction [20], Lianhua Qingke granule [21], Reyanning mixture [22], Jiawei Dayuan Decoction [24], diammonium glycyrrhizinate [26].

The frequency of each Chinese herb in this meta-analysis was also summarized manually. The top 3 ranked Chinese herbs were honeysuckle (58.33%) [15–19, 23, 25], forsythia (50.00%) [15–17, 19, 21, 23], and ephedra (50.00%) [15, 19–21, 23, 24].

Four dosage formulations of oral CHM were included, including granule [15–17, 21, 23, 24], oral liquid [18, 22, 25], capsule [19, 26], and decoction [20]. The most commonly used dosage formulation was granule (50.00%) [15–17, 21, 23, 24].

**Table 1. The characteristics of included RCTs.**

| First author | Type of COVID-19 | Sample size (M/F) | Age (yrs) | Intervention | Control | Duration | Outcome measures |
|---|---|---|---|---|---|---|---|
| Duan C [15] | mild | T:82(39/43) C:41 (23/18) | T:51.99±13.88 C:50.29±13.17 | Jinhua Qinggan granule and conventional therapy | Conventional therapy | 5 days | ⑤+⑦ |
| Fu [16] | mild/ moderate | T:32(17/15) C:33 (19/14) | T:43.26±7.15 C:43.68±6.45 | Toujie Quwen granule and conventional therapy | Conventional therapy | 10 days | ①+②+③+⑤+⑥+⑦ |
| Fu XX [17] | moderate | T:37(19/18) C:36 (19/17) | T:45.26±7.25 C:44.68±7.45 | Toujie Quwen granule and conventional therapy | Conventional therapy | 15 days | ②+③+⑤+⑥+⑦ |
| Hu F [18] | moderate | T:100(49/51) C:100(55/45) | T:47.00±14.06 C:49.28±11.14 | Jinyinhua oral liquid and conventional therapy | Conventional therapy | 10 days | ①+③+④+⑦ |
| Hu K [19] | mild/ moderate | T:142(79/63) C:142(71/71) | T:50.4±15.2 C:51.8 ±14.8 | Lianhua Qingwen capsule and conventional therapy | Conventional therapy | 14 days | ①+②+③+④+⑤+⑦ |
| Qiu M [20] | moderate | T:25(13/12) C:25 (14/11) | T:53.35±18.35 C:51.32±14.62 | Maxing Xuanfei Jiedu Decoction and conventional therapy | Conventional therapy | 10 days | ①+③+⑤ |
| Sun HM [21] | mild/ moderate | T:32(17/15) C:25 (11/14) | T:45.4±14.10 C:42.0±11.70 | Lianhua Qingke granule and conventional therapy | Conventional therapy | 14days | ①+③+⑤ |
| Yang MB [22] | moderate | T:26(16/10) C:23 (9/14) | T:50.35±13.37 C:47.17±16.57 | Reyanning mixture and conventional therapy | Conventional therapy | 7 days | ③+④+⑤+⑥+⑦ |
| Yu P [23] | mild/ moderate | T:147(82/65) C:148(89/59) | T:48.27±9.56 C:47.25±8.67 | Lianhua Qingwen granule and conventional therapy | Conventional therapy | 7 days | ①+②+③+⑤+⑥+⑦ |
| Zhang CT [24] | moderate | T:22(9/13) C:23 (10/13) | T:53.7±3.5 C: 55.6 ±4.2 | Jiawei Dayuan Decoction and conventional therapy | Conventional therapy | 7 days | ①+⑤+⑥+⑦ |
| Zhang YL [25] | moderate | T:80(50/30) C:40 (23/17) | T:53.4±13.70 C:52.0±14.10 | Jinyinhua oral liquid and conventional therapy | Conventional therapy | 10 days | ③+⑤+⑦ |
| Zhou WM [26] | moderate | T:52(32/20) C:52 (28/24) | T:52.47±10.99 C:51.11±9.87 | diammonium glycyrrhizinate and conventional therapy | Conventional therapy | 14 days | ②+⑥+⑦ |

①: Lung CT; ②: Clinical cure rate; ③: Rate of conversion to severe cases; ④: Virus nucleic acid testing; ⑤: Clinical symptoms; ⑥: Inflammatory biomarkers; ⑦: Adverse events.

### Efficacy and safety assessment

**Clinical efficacy.** *Lung CT.* The evaluation criteria for a lung CT refer to the COVID-19 Guidelines for Imaging Assisted Diagnosis [27]. Lung CT can evaluate the curative effect through the parameters basic absorption, improvement, no change, and aggravation. If the lesion range disappears ≥70%, it indicates basic absorption. If the lesion range disappeared ≥30%, it indicates improvement. If there was no change in the lesion range, it indicates no change. If the extent of the lesion increased by ≥30%, it indicates aggravation. The effectiveness of therapy based on lung CT = (basic absorption cases + improvement cases)/total cases × 100%. Seven trials enrolling 845 patients mentioned lung CT [16, 19–24]. A fixed-effects model was used due to no significant heterogeneity ($I^2$ = 8%, $P$ = 0.37). Meta-analysis revealed that combination therapy could significantly improve lung CT [RR = 1.26, 95%CI (1.15, 1.38), $P$<0.00001] (Fig 3a). Subgroup analysis showed that there was a significant difference between subgroups with 7 days of treatment duration ($P$ = 0.03) and 10 to 14 days of treatment duration ($P$<0.00001) (Fig 3a).

*Clinical cure rate.* Clinical cure standards refer to Guiding Principles for Clinical Research of New Chinese Materia Medica [28]. Therapeutic effects are classified as effective, improved, and ineffective. If the TCM symptom score is reduced by more than 70%, it suggests effectiveness. If the TCM symptom score is reduced by more than 30%, it represents improved symptoms. If the TCM symptom score is reduced by less than 30%, it represents ineffective treatment. Clinical cure rate = (effective cases + improved cases)/total cases × 100%. Five trials

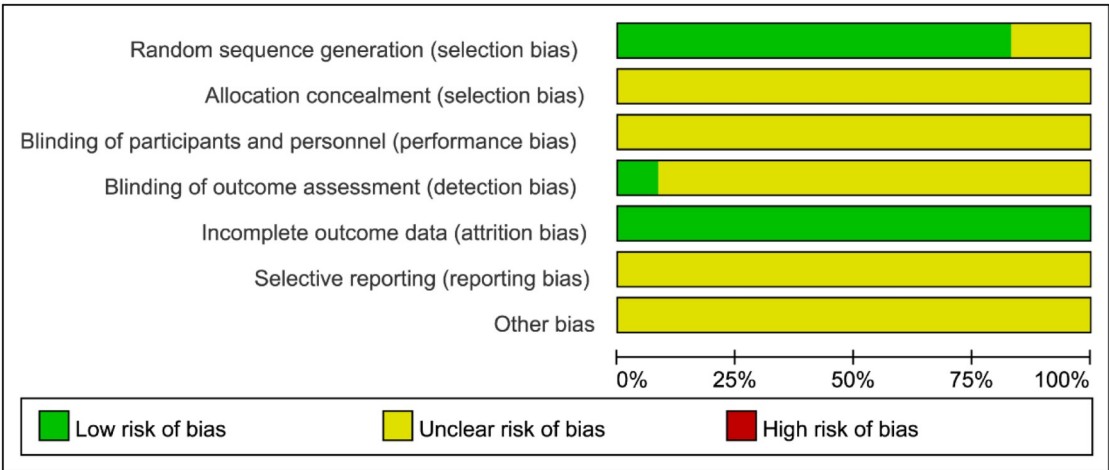

(a) Risk of bias graph.

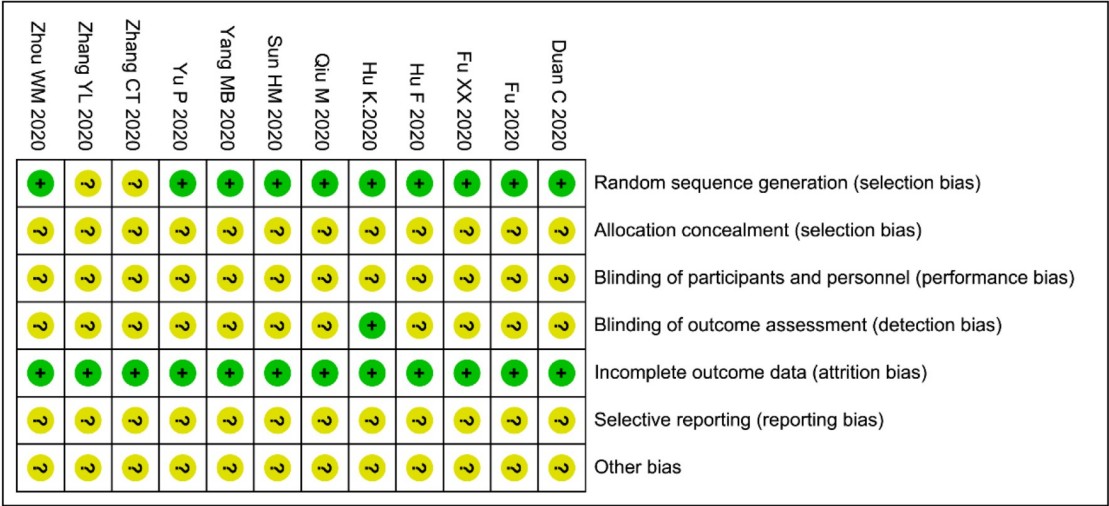

(b) Risk of bias summary.

**Fig 2. Assessment of methodological quality.** (a) Risk of bias graph. (b) Risk of bias summary.

enrolling 821 participants reported clinical cure rate [16, 17, 19, 22, 26]. A fixed-effects model was used due to no significant heterogeneity ($I^2 = 0\%$, $P = 0.77$). The outcome indicated clinical cure rate in combination therapy was higher than conventional therapy [RR = 1.26, 95%CI (1.16, 1.38), $P<0.00001$] (Fig 3b).

*Rate of conversion to severe cases.* Nine trials enrolling 1121 patients reported rate of conversion to severe cases [16–23, 25]. A fixed-effects model was used due to no significant heterogeneity ($I^2 = 0\%$, $P = 0.83$). The results showed that combination therapy could significantly reduce rate of conversion to severe cases [RR = 0.48, 95%CI (0.32, 0.73), $P = 0.0005$] (Fig 3c).

*Viral nucleic acid testing.* Negative rate of viral nucleic acid testing = (negative cases at the end of the trial − negative cases before the trial)/total cases × 100%. Four trials enrolling 581 patients reported viral nucleic acid testing [18–19, 22, 25]. A random-effects model was used due to the significant heterogeneity ($I^2 = 57\%$, $P = 0.08$). Meta-analyses revealed no statistical difference in viral nucleic acid testing [RR = 1.09, 95%CI (0.98, 1.21), $P = 0.13$] (Fig 3d).

**Table 2. The components of CHM.**

| References | CHM | Components |
|---|---|---|
| Duan C [15] | Jinhua Qinggan granule | Jinyinhua 10g, Shigao 10g, Mahuang (processed with honey) 10g, Kuxingren (stir-frying) 10g, Huangqin 10g, Lianqiao 10g, Zhebeimu 10g, Zhimu 10g, Niubangzi 10g, Qinghao 10g, Bohe 10g, Gancao10g |
| Fu [16] | Toujie Quwen granule | Lianqiao 30g, Shancigu 20g, Jinyinhua 15g, Huangqin 10g, Daqingye 10g, Chaihu 5g, Qinghao 10g, Chantui 10g, Qianhu 5g, Chuanbeimu 10g, Zhebeimu 10g, Wumei 30g, Xuanshen 10g, Huangqi 45g, Fuling 30g, Taizishen 15g |
| Fu XX [17] | Toujie Quwen granule | Lianqiao 30g, Shancigu 20g, Jinyinhua 15g, Huangqin 10g, Daqingye 10g, Chaihu 5g, Qinghao 10g, Chantui 10g, Qianhu 5g, Chuanbeimu 10g, Zhebeimu 10g, Wumei 30g, Xuanshen 10g, Huangqi 45g, Fuling 30g, Taizishen 15g |
| Hu F [18] | Jinyinhua oral liquid | Jinyinhua 5.4g |
| Hu K [19] | Lianhua Qingwen capsule | Lianqiao, Jinyinhua, Mahuang (stir-frying), Kuxingren (stir-frying), Shigao, Banlangen, Guanzhong, Yuxingcao, Huoxiang, Dahuang, Hongjingtian, Bohe, Gancao |
| Qiu M [20] | Maxing Xuanfei Jiedu Decoction | Mahuang 9g, Kuxingren 12g, Shigao 15~30g, Zhebeimu 12g, Chantui 10g, Jiangchan 15g, Jianghuang 12g, Jiegeng 12g, Zhiqiao 12g, Caoguo 9g, Caodoukou 12g |
| Sun HM [21] | Lianhua Qingke granule | Mahuang, Sangbaipi, Kuxingren (stir-frying), Lianqiao, mountain honeysuckle, Dahuang |
| Yang MB [22] | Reyanning mixture | Pugongying, Huzhang, Baijiang Herba cum Radice, Banzhilian |
| Yu P [23] | Lianhua Qingwen granule | Lianqiao, Jinyinhua, Mahuang (stir-frying), Kuxingren (stir-frying), Shigao, Banlangen, Guanzhong, Yuxingcao, Huoxiang, Dahuang, Hongjingtian, Bohe, Gancao |
| Zhang CT [24] | Jiawei Dayuan Decoction | Mahuang (stir-frying) 10g, Xingren 15g, crude gypsum 20g, trichosanthes bark 20g, Dahuang (Stir-fry with yellow rice wine) 6g, Tinglizi 10g, Taoren 10g, Caoguo 6g, Binglang 10g, Cangzhu 10g |
| Zhang YL [25] | Jinyinhua oral liquid | Jinyinhua 5.4g |
| Zhou WM [26] | diamine glycyrrhizinate | diamine glycyrrhizinate |

**Clinical symptoms.** *Fever.* Three trials enrolling 205 patients mentioned fever reduction cases [15, 21, 25]. A random-effects model was used due to the significant heterogeneity ($I^2$ = 95%, $P < 0.00001$). Meta-analysis showed that there was no statistical difference on fever reduction cases [RR = 1.14, 95%CI (0.58, 2.25), $P$ = 0.70] (Fig 4a). Four trials involved 482 participants reported TCM symptom score of fever [16, 17, 22, 23]. A random-effects model was used due to the significant heterogeneity ($I^2$ = 79%, $P$ = 0.009). The pooled result showed that combination therapy could result in a significant reduction in TCM symptom score of fever [MD = -0.62, 95%CI (-0.79, -0.45), $P < 0.00001$] (Fig 4b).

*Cough.* Three trials enrolling 205 patients mentioned cough reduction cases [15, 21, 25]. A fixed-effects model was used due to no significant heterogeneity ($I^2$ = 0%, $P$ = 0.89). Meta-analyses revealed that combination therapy could significantly reduce cough cases [RR = 1.43, 95%CI (1.16, 1.75), $P$ = 0.0006] (Fig 4c). Four trials enrolling 482 participants reported TCM symptom score of fever [16, 17, 22, 23]. A random-effects model was used due to the significant heterogeneity ($I^2$ = 84%, $P$ = 0.0003). The pooled estimate found combination therapy decreased TCM symptom score of cough [MD = -1.07, 95%CI (-1.29, -0.85), $P < 0.00001$] (Fig 4d).

*Fatigue.* Three trials enrolling 205 patients mentioned fatigue reduction cases [15, 21, 25]. A fixed-effects model was used due to no significant heterogeneity ($I^2$ = 28%, $P$ = 0.25). Meta-

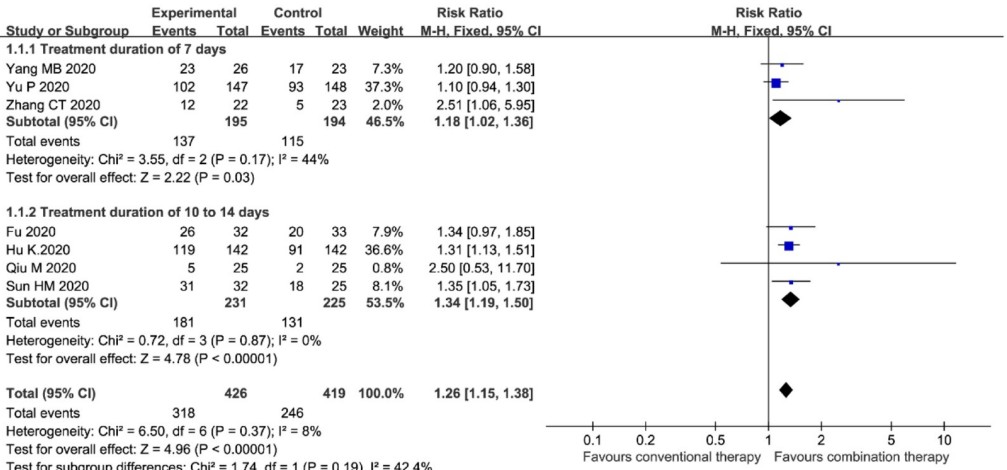

(a) Lung CT.

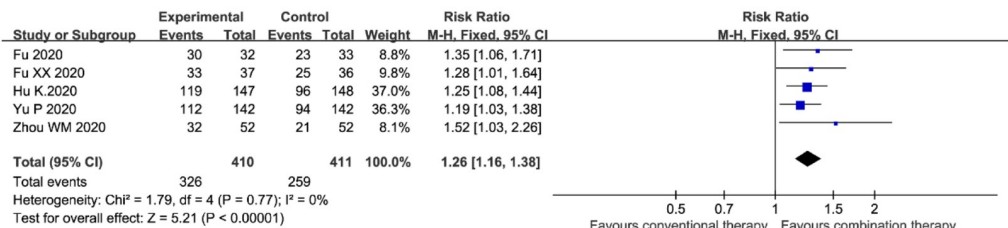

(b) Clinical cure rate.

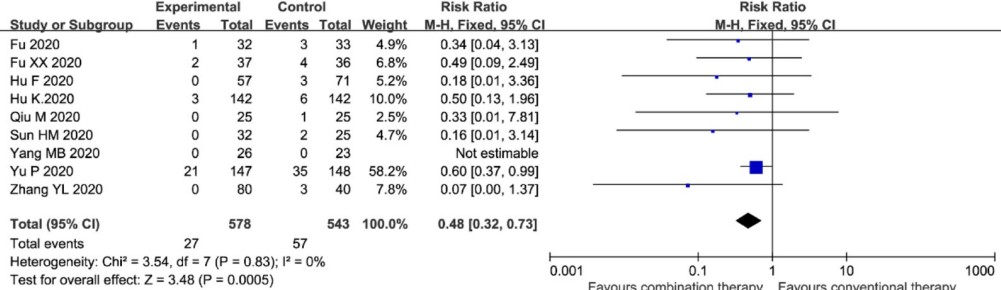

(c) Rate of conversion to severe cases.

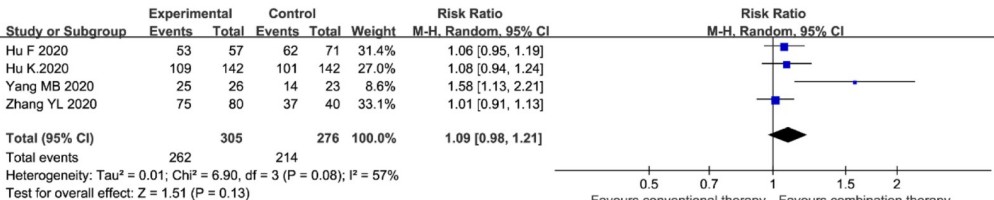

(d) Viral nucleic acid testing.

**Fig 3.** Forest plot of the effects of combination therapy for outcomes of (a) lung CT, (b) clinical cure rate, (c) rate of conversion to severe cases, (d) viral nucleic acid testing.

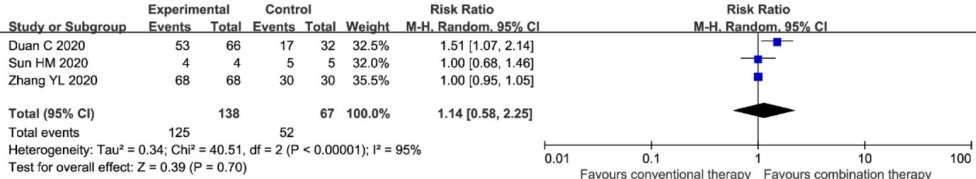

(a) Fever reduction cases.

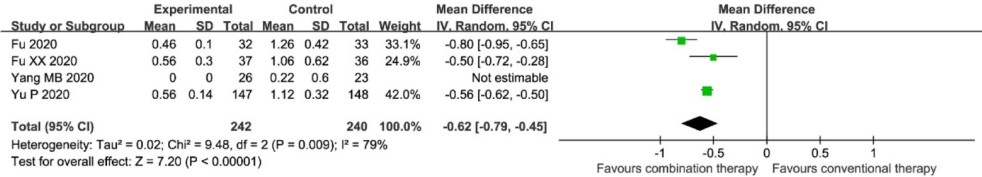

(b) TCM symptom score of fever.

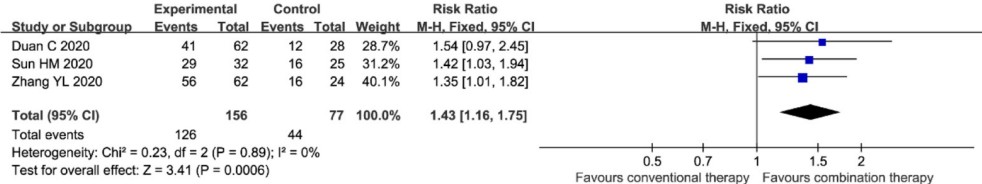

(c) Cough reduction cases.

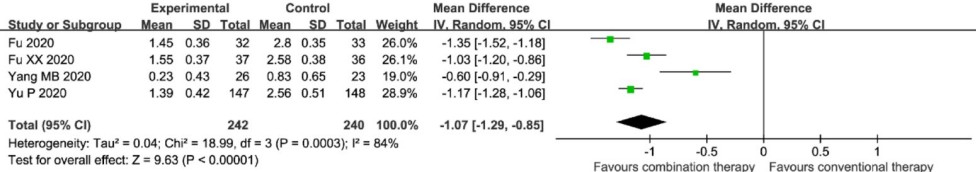

(d) TCM symptom score of cough.

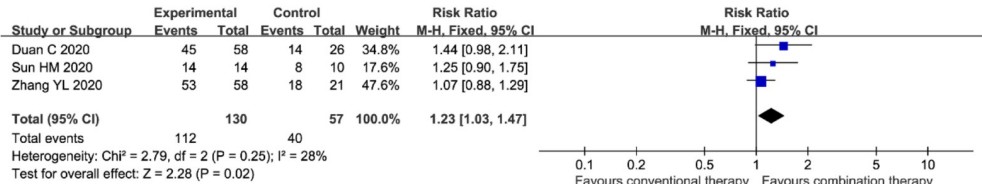

(e) Fatigue reduction cases.

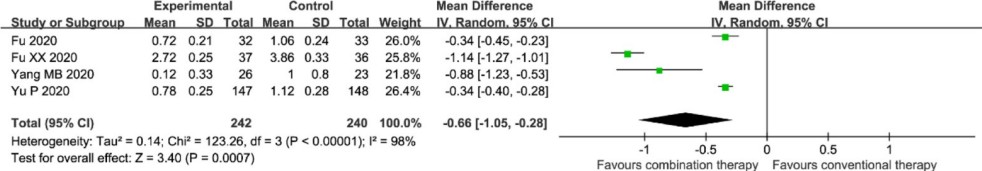

(f) TCM symptom score of fatigue.

**Fig 4.** Forest plot of the effects of combination therapy for outcomes of (a) fever reduction cases, (b) TCM symptom score of fever, (c) cough reduction cases, (d) TCM symptom score of cough, (e) fatigue reduction cases, (f) TCM symptom score of fatigue.

analysis showed that combination therapy could significantly reduce fatigue cases [RR = 1.23, 95%CI (1.03, 1.47), $P$ = 0.02] (Fig 4e). Four trials enrolling 482 participants reported TCM symptom score of fever [16, 17, 22, 23]. A random-effects model was used due to the significant heterogeneity ($I^2$ = 98%, $P$<0.00001). The pooled result found combination therapy decreased TCM symptom score of fatigue [MD = -0.66, 95%CI (-1.05, -0.28), $P$ = 0.0007] (Fig 4f).

**Inflammatory biomarkers.** *WBC count.* Four trials enrolling 478 participants mentioned WBC count [16, 17, 23, 24]. A fixed-effects model was used due to no significant heterogeneity ($I^2$ = 5%, $P$ = 0.37). Meta-analysis revealed that combination therapy could significantly increase WBC count [MD = 0.38, 95%CI (0.31, 0.44), $P$<0.00001] (Fig 5a). Subgroup analysis showed that there was a significant difference between subgroups with 7 days of treatment duration ($P$<0.00001) and 10 to 15 days of treatment duration ($P$<0.00001) (Fig 5a).

*LYM count.* Four trials enrolling 482 participants reported LYM count [16, 17, 22, 23]. A random-effects model was used due to the significant heterogeneity ($I^2$ = 97%, $P$<0.00001). The pooled estimate found combination therapy increased LYM count [MD = 0.26, 95%CI (0.05, 0.47), $P$ = 0.01] (Fig 5b). Subgroup analysis showed that there was a significant difference between subgroups with 7 days of treatment duration ($P$<0.00001) and 10 to 15 days of treatment duration ($P$ = 0.0002) (Fig 5b).

*CRP.* Six trials enrolling 631 participants reported CRP [16, 17, 22–24, 26]. A random-effects model was used due to the significant heterogeneity ($I^2$ = 96%, $P$<0.00001). The pooled result found combination therapy decreased CRP [MD = -5.46, 95%CI (-8.19, -2.72), $P$<0.0001] (Fig 5c). Subgroup analysis showed that there was a significant difference between subgroups with 7 days of treatment duration ($P$ = 0.02) and 10 to 15 days of treatment duration ($P$ = 0.04) (Fig 5c).

**Adverse drug events.** *Total number of adverse drug events cases.* Ten trials enrolling 1286 participants reported total number of adverse drug events cases [15–19, 22–26]. A random-effects model was used due to the significant heterogeneity ($I^2$ = 63%, $P$ = 0.03). Meta-analyses revealed no statistical difference in total number of adverse drug events cases [RR = 1.13, 95% CI (0.45, 2.83), $P$ = 0.79] (Fig 6a).

*Nausea and vomiting.* Two trials enrolling 388 participants reported nausea and vomiting [19, 26]. A fixed-effects model was used due to no significant heterogeneity ($I^2$ = 0%, $P$ = 0.74). Subgroup analysis suggested no statistical difference in nausea and vomiting [RR = 1.09, 95% CI (0.49, 2.41), $P$ = 0.83] (Fig 6b).

*Diarrhea.* Five trials enrolling 759 participants reported total number of adverse drug events cases [15, 18, 19, 25, 26]. A random-effects model was used due to the significant heterogeneity ($I^2$ = 70%, $P$ = 0.009). Subgroup analysis showed no statistical difference in diarrhea [RR = 1.72, 95%CI (0.34, 8.67), $P$ = 0.51] (Fig 6c).

*Abnormal liver function.* Two trials enrolling 388 participants reported total number of adverse drug events cases [19, 26]. A random-effects model was used due to the significant heterogeneity ($I^2$ = 78%, $P$ = 0.03). Subgroup analysis revealed no statistical difference in abnormal liver function [RR = 0.41, 95%CI (0.05, 3.69), $P$ = 0.43] (Fig 6d).

One trial reported that there were 8 cases of poor appetite, 1 case of headache, and 8 cases of renal dysfunction in combination therapy group [19].

## Sensitivity analysis

Sensitivity analysis showed that there was a small change in the effect amount, and a significant difference in lung CT, clinical cure rate, rate of conversion to severe cases, TCM symptom score of fever, cough reduction cases, TCM symptom score of cough, TCM symptom score of fatigue, WBC count, and CRP, which indicated the above meta-analysis results to be robust.

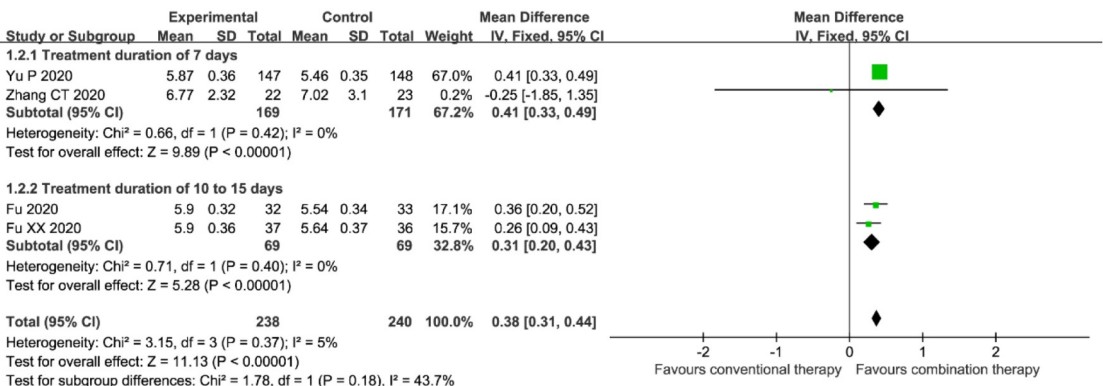

(a) WBC count.

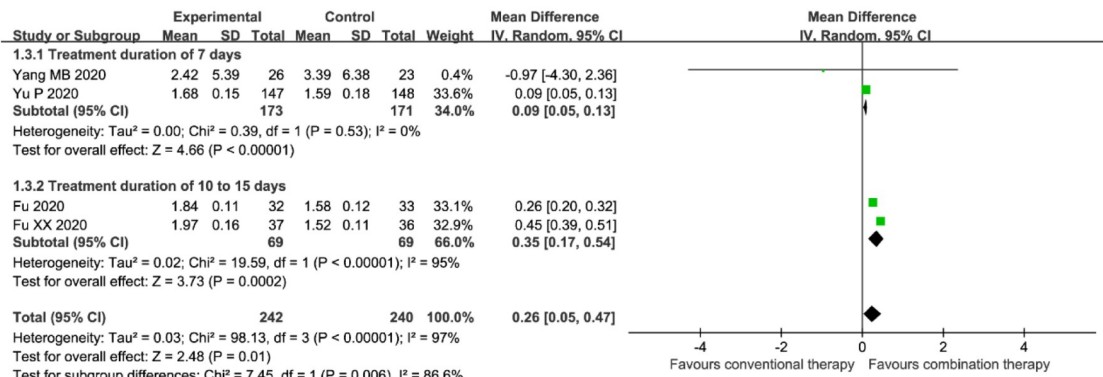

(b) LYM count.

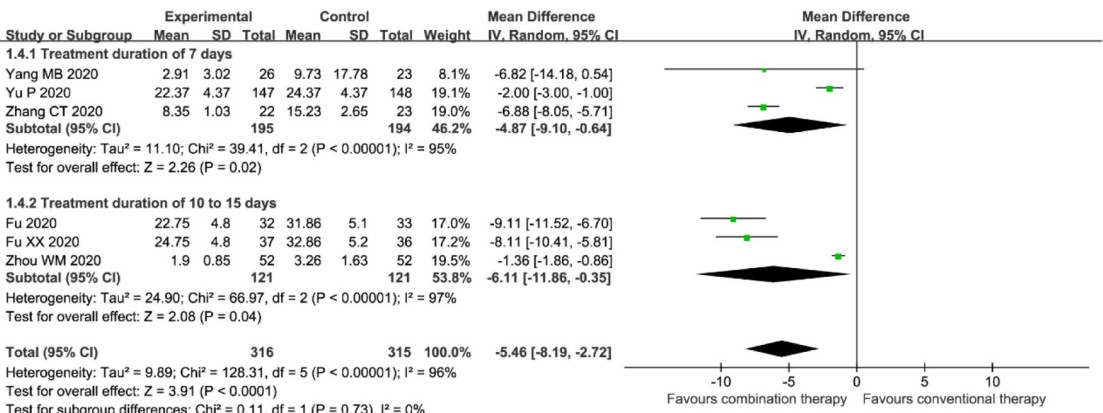

(c) CRP.

**Fig 5.** Forest plot of the effects of combination therapy for outcomes of (a) WBC count, (b) LYM count, (c) CRP.

## Publication bias

In our study, ten trials reported adverse drug events [15–19, 22–26]. Among them, five trials reported that no adverse drug events were observed [16, 17, 22–24]. The funnel plot was used to analyze the reported adverse events trials to explore the bias (Fig 7). The funnel plot is symmetrical, indicating no obvious deviation.

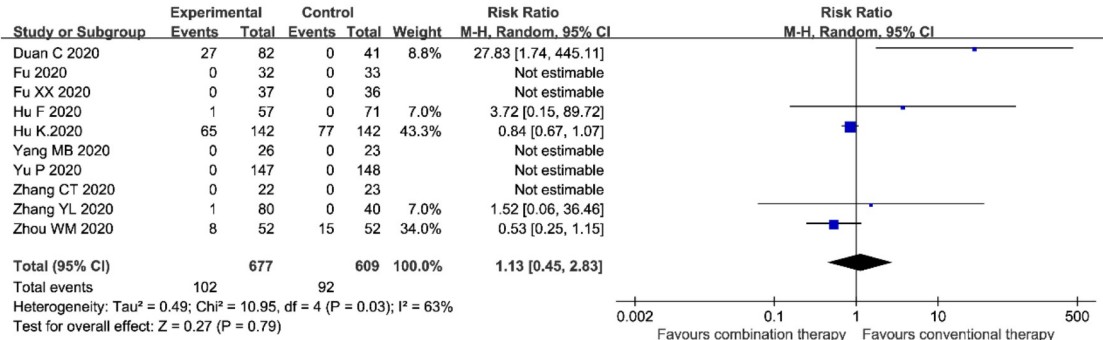

(a) Total number of adverse drug events cases.

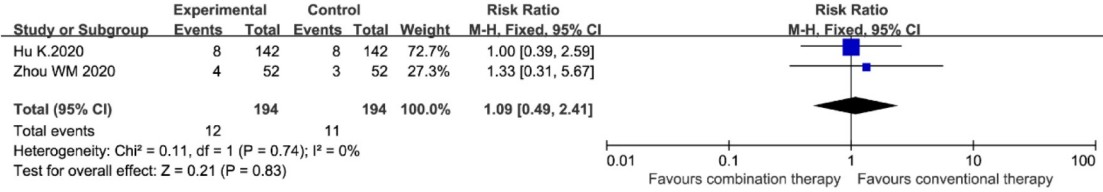

(b) Nausea and vomiting.

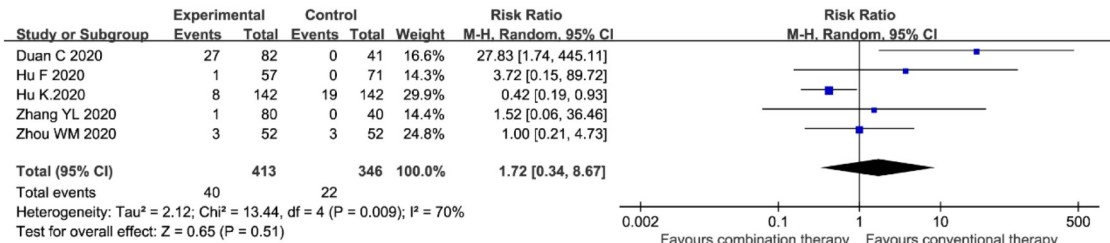

(c) Diarrhea.

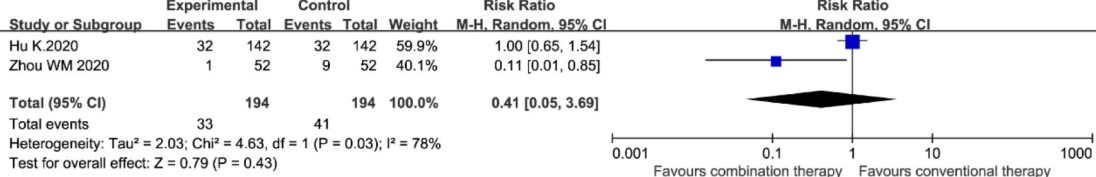

(d) Abnormal liver function.

**Fig 6.** Forest plot of the safety of combination therapy for outcomes of (a) total number of adverse drug events cases, (b) nausea and vomiting, (c) diarrhea, (d) abnormal liver function.

## Discussion

The clinical classification of COVID-19 is mild, moderate, severe, and critical [7]. Severe COVID-19 is more likely to have serious complications, such as shock, acute respiratory distress syndrome (ARDS), arrhythmia, and acute heart injury [29, 30], all of which significantly increase the difficulty and cost of treatment. Therefore, it is of great significance to prevent COVID-19 from developing from mild or moderate to severe. In our study, it was found that compared with conventional therapy alone, mild to moderate COVID-19 patients treated with combination therapy of CHM and conventional therapy had more benefit. Similar studies

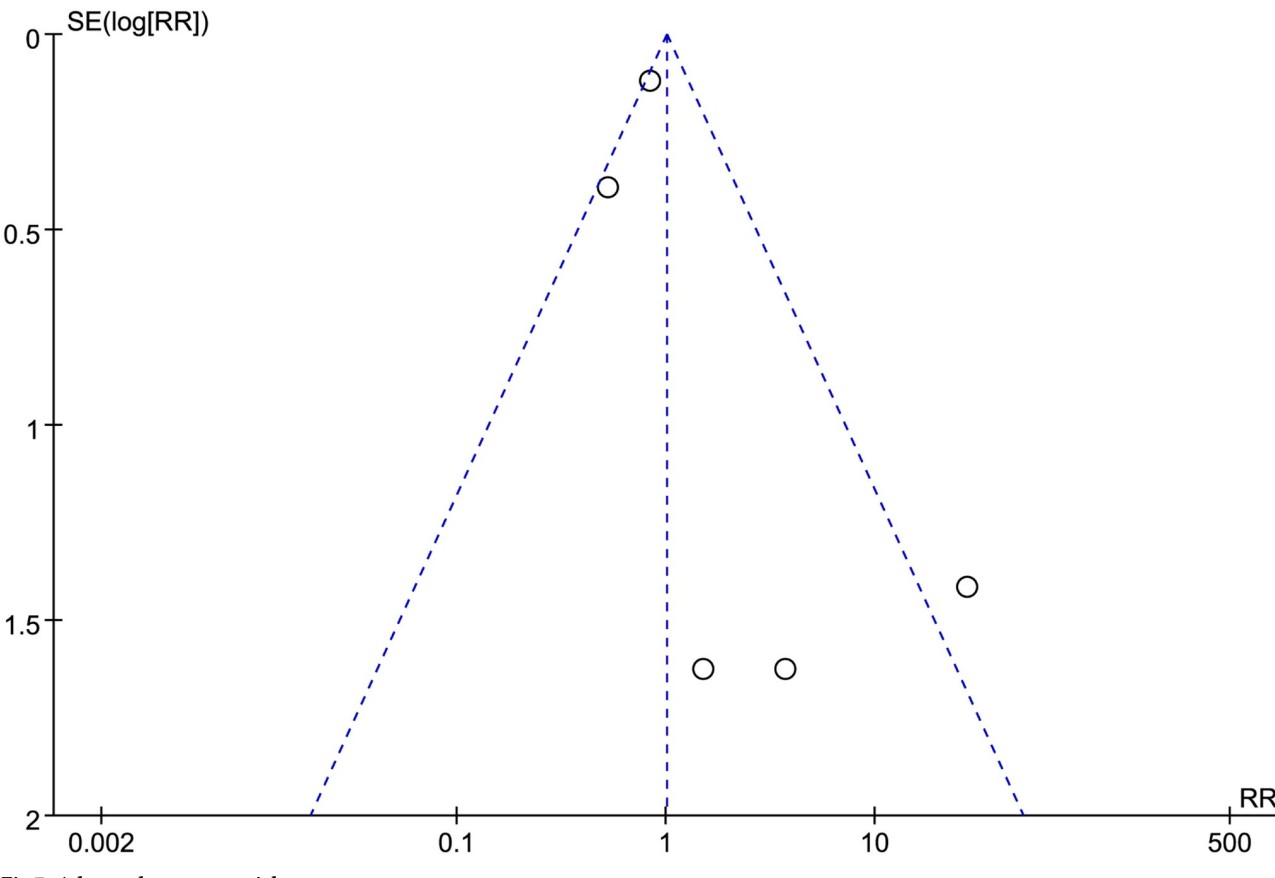

**Fig 7. Adverse drug events trials.**

have shown that CHM has positive effects in COVID-19 patients [31–33]. Facing such a severe COVID-19 epidemic, Western countries should pay attention to the therapeutic effect of CHM for COVID-19.

According to the theory of traditional Chinese medicine (TCM), epidemic disease refers to an acute infectious disease characterized by sudden onset, rapid transmission, dangerous conditions, and strong infectivity after feeling pestilence evil [34]. COVID-19 is an "epidemic disease" of TCM in light of its incidence mode and epidemic trend [7]. The pathogenesis of mild to moderate COVID-19 is dampness-heat or cold-dampness obstructing the lung [7]. Therefore, CHM, with the effect of clearing heat, eliminating dampness, resolving phlegm, and dispersing cold, is widely used [7]. In the included studies, nine different oral CHM were used, including Lianhua Qingwen capsules and granules, Toujie Quwen granules, Jinyinhua oral liquids, diammonium glycyrrhizinate, etc. Lianhua Qingwen capsules originate from classical Chinese herbal formulas and can decrease influenza A virus (H1N1) replication, lung lesions, and inflammation [35]. Additionally, Lianhua Qingwen capsules may reduce lung injury and help eliminate SARS-CoV-2 infection by regulating Akt1 [36]. One study has shown that Toujie Quwen granules may have therapeutic effects on COVID-19 by regulating SARS-CoV-2 infection, immune and inflammation-related targets, and pathways [37]. Diammonium glycyrrhizinate is used as a hepatic protector and is the main component of licorice root extracts [38]. Diammonium glycyrrhizinate can decrease serum ALT and AST levels, improve histological damage, downregulate inflammatory cytokines, and inhibit the apoptosis of T lymphocytes in the thymus [38].

Among the nine oral CHM, the most frequently used Chinese herb was honeysuckle, followed by forsythia and ephedra. Honeysuckle and forsythia have the function of clearing heat toxicity and dispersing wind heat in the theory of TCM [5]. Honeysuckle polysaccharide is an active component of honeysuckle that can regulate nonspecific immunity [39], inhibit the expression of the inflammatory factors TNF-α and IL-1β [40], and inhibit a variety of viruses [41]. Phillyrin is an active component of forsythia that has antiviral and anti-inflammatory activities [42, 43]. Ephedra has the function of dissipating cold and diffusing the lung to calm panting in TCM theory [5]. Ephedrine is an active component of ephedra that can increase the production of the anti-inflammatory cytokine IL-10, reduce the production of the proinflammatory cytokines TNF-α and IL-12 [44], and play an antiviral role by inhibiting viral replication [45].

Mild to moderate COVID-19 patients treated with combination therapy of CHM and conventional therapy had better outcomes in parameters including clinical efficacy, clinical symptoms, and inflammatory response. Our study found that compared with conventional therapy alone, combination therapy could improve the scores of symptoms such as fever, cough, and fatigue and reduce cough cases. Combination therapy could increase WBC count and decrease CRP. This is related to the fact that CHM can improve the host immune response and downregulate inflammatory cytokines [35, 38, 46]. Immunopathological changes, including relatively lower levels of WBCs and LYMs and markedly higher levels of CRP and inflammatory cytokines, are correlated with COVID-19 severity [47, 48]. Immune suppression and inflammatory injury are also important drivers of COVID-19 progression [49]. Cytokine storm is a hyperproduction of inflammatory cytokines, which can lead to ARDS aggravation and widespread tissue damage resulting in acute lung injury, multiorgan failure and death [50, 51]. Targeting cytokines during the management of COVID-19 patients could improve survival rates [51]. In our study, we also found that combination therapy had a better effect on improving lung CT parameters, promoting the clinical cure rate, and reducing the rate of conversion to severe cases.

Due to different formulations and unclear compositions, CHM has many unknown factors to be solved. In our study, we found that CHM formulations used in the combination therapy group were different, and the quality of herbal intervention was unclear. CHM is likely to require a standard treatment. In addition, the quality of herbal formulas should be monitored through standardization. In this way, the best evidence can be systematically summarized to better provide an evidence-based basis for TCM decision-making. CHM treatment, which is based on individualized assessment, can be affected by different diet practices and weather, resulting in its difficulty of use in Western countries. Therefore, we think it is necessary for Western countries to hire TCM experts to participate in the treatment of COVID-19. Safety issues should be a concern when CHM is used for COVID-19. In our study, we found that most of the included trials reported adverse drug events. Combination therapy did not increase adverse drug events. The funnel plot of adverse drug events indicated no obvious deviation.

However, it was a common problem that most of the included trials had poor methodological design and that the merger statistical analysis of some outcomes had unexplained heterogeneity. More high-quality trials are needed in the future. Despite the poor methodology and unexplained heterogeneity, our findings are very valuable and timely in view of the lack of specific drugs approved for COVID-19.

## Limitations

Despite the usefulness of our findings, this review also has several limitations that could be improved upon in future studies. First, most of the included trials had deficiencies in

methodology design, including hidden allocation and inadequate reporting of blind methods. Second, the composition, dosage, and frequency of CHM were different in the treatment groups. Third, the multicenter trials were lacking. In addition, the duration of the included trials ranged from 5 to 15 days. Therefore, it is necessary to design more high-quality trials with a multicenter, larger sample size, and longer follow-up to better observe the efficacy and possible adverse events of CHM combined with conventional therapy in the treatment of mild to moderate COVID-19.

## Conclusion

Chinese herbal medicine combined with conventional therapy could be effective and safe in the treatment of mild to moderate COVID-19. Combination therapy can improve the clinical cure rate, main clinical symptoms, imaging and laboratory indexes, and reduce the rate of conversion to severe cases. However, because COVID-19 is a sudden disease, it is difficult to carry out double-blind clinical trials, which leads to insufficient methodology in the existing related trials. Therefore, more high-quality trials are needed to evaluate the efficacy and safety of Chinese herbal medicine combined with conventional therapy in the treatment of adults with mild to moderate COVID-19 in the future.

## Supporting information

**S1 Checklist. PRISMA 2009 checklist.**
(DOC)

## Author Contributions

**Conceptualization:** Lipeng Shi, Wenfu Cao.

**Data curation:** Xuqin Du, Lipeng Shi.

**Formal analysis:** Xuqin Du.

**Funding acquisition:** Xuqin Du.

**Investigation:** Xuqin Du, Lipeng Shi.

**Methodology:** Xuqin Du, Lipeng Shi.

**Software:** Xuqin Du.

**Supervision:** Wenfu Cao.

**Validation:** Biao Zuo, Aimin Zhou.

**Writing – original draft:** Xuqin Du.

**Writing – review & editing:** Xuqin Du, Lipeng Shi, Biao Zuo, Aimin Zhou.

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
