## [Decision Letter · Decision Letter 0]

15 Feb 2021

PONE-D-20-38124

Chinese herbal medicine in adults with mild to moderate coronavirus disease 2019(COVID-19): A systematic review and meta-analysis

PLOS ONE

Dear Dr. Shi,

Thank you for submitting your manuscript to PLOS ONE. After careful consideration, we feel that it has merit but does not fully meet PLOS ONE’s publication criteria as it currently stands. Therefore, we invite you to submit a revised version of the manuscript that addresses the points raised during the review process.

We look forward to receiving your revised manuscript.

Kind regards,

Ahmed Negida, MD

Academic Editor

PLOS ONE

Journal Requirements:

2. Please include your tables as part of your main manuscript and remove the individual files. Please note that supplementary tables should remain uploaded as separate "supporting information" files.

Reviewers' comments:

Reviewer's Responses to Questions

**Comments to the Author**

1. Is the manuscript technically sound, and do the data support the conclusions?

Reviewer #1: Yes

Reviewer #2: Partly

Reviewer #3: Partly

Reviewer #4: Yes

2. Has the statistical analysis been performed appropriately and rigorously? 

Reviewer #1: Yes

Reviewer #2: I Don't Know

Reviewer #3: Yes

Reviewer #4: Yes

3. Have the authors made all data underlying the findings in their manuscript fully available?

Reviewer #1: Yes

Reviewer #2: No

Reviewer #3: Yes

Reviewer #4: Yes

4. Is the manuscript presented in an intelligible fashion and written in standard English?

Reviewer #1: Yes

Reviewer #2: No

Reviewer #3: No

Reviewer #4: Yes

5. Review Comments to the Author

Reviewer #1: kindly, find the primary review and comments in the attached Pdf file, in addition, please pay attention to the following points:

-The main claim of the paper is clear and significant, specially in such unprecedent situation.

-The analysis o data supports the claim of the paper, however; it would be better to connect this study with more previous published data and literatures in a way that reduce duplication and support the findings of this paper.

-a more detailed protocol of the statistical analysis is needed especially, most of the data used in the analysis has been retrieved from papers in Chinese language.

-Type of samples in treatment and control groups doesn't exclude the possibility of synergistic/ combination effect between CHM and western medicine. have you had any studies that used CHM only on separate groups as a treatment? Was there any control group that didn't receive any treatment? is there any information about hospitalization or receiving any other special care(ex. ventilator) beside the treatment?

i.e: we can't conclude for sure the CHM as a separate, effective, and safe treatment for mild to moderate COVID-19.

Reviewer #2: Valuable data was provided in this manuscript, which are not easily assessible for international readers outside China. Hence, I have to stress that this manuscript presents precious and valuable data that will benefit the literature and improve understanding of the role of TCM in COVID-19. However, in general, I find that there is lack of clarity in definition of many things including outcome measures and treatment groups. Importantly, the discussion was superficial. There needs to be correlation between ROB, quality of study, heterogeneity and interpretation of results. Please find my suggestion as below and as specify in the attachment:

1. Strongly suggest for professional language/ scientific proof-reading to correct grammar, sentence structuring, and selection of words that are preferred to represent precise scientific writing for the entire manuscript. Kindly check for the use of oxford comma and appropriate/excessive use of connective words throughout. The authors in particular like to start sentences with the word "And". Spacing between words and symbols needs to be checked and made consistent.

2. The eligibility criteria can be rewritten as inclusion and exclusion criteria clearly; or rearrange with clearer subtopics differentiation. The different levels of the subtopics in the methods needs to be clear. For example (here I am using numbers to explain an example of how the different levels needs to be clarified. It is to the authors discretion on presenting this without the numbers)

2.0 Methods

2.1 Eligibility criteria

2.1.1 Type of studies

2.1.2 Participant characteristics

2.2 Literature search

2.2.1 Search strategy

2.2.2 Study selection and data extraction

2.3 Data analysis

2.3.1Methodological quality assessment

2.3.2 Meta-analyses

3. Specific to the methods

a. Kindly check against the PRISMA checklist- Present full electronic search strategy for at least one database (please present the combination of keywords used); Describe method of data extraction from reports (e.g., piloted forms, independently, in duplicate) and any processes for obtaining and confirming data from investigators (kindly mention if attempts were made to seek for additional data)

b. Clarify inclusion criteria- oral Chinese herbal medicine only

c. Outcome measures need to be well defined e.g. what is clinical cure rate, what is effective rate of lung CT

4. Results

a. arrange the level of subheadings accordingly as suggested for methods

b. definition of CHM and CWM needs to be clear- the naming of the groups. Although it is mentioned that CHM group received both herbal and western medicine in methods, CHM is still abbreviated as chinese herbal medicine. The results are mostly written as 'the outcomes are better with treatment by CHM', which can be confusing to interpret, and easily misunderstood as if CHM solely (without western medicine) is beneficial. Suggest to clearly describe what each group means with distinct abbreviations for groups. Perhaps it is also because of the choice of word 'by' which when read, is interpreted this way, hence consider rewriting the results section with more precise selection of words.

5. Discussion

Although an interesting topic with very valuable data (I cannot emphasize this enough, this is very valuable data), the discussion is superficial and lacked depth. few suggestion of topics to discuss include

- heterogeneity of the studies and the impact on the findings.

- impact of different formulations used and how did the authors came to collectively interpreting them in the same meta-analyses (also consider that different herbs would have acted differently, and certainly herb-herb interaction should be discussed)

- risk of bias and how that affects results interpretation

- discuss on adverse events, reporting bias?

- quality of herbal intervention used

- suggest to consider consort checklist for tcm to evaluate quality of reporting which can further strengthen discussion

- how does this new information applies to the global scenario and what are the challenges of applying TCM in this scenario

- difference between TCM approach (Which is based on individualised assessment, and can be even affected by factors such as diet, body type, environment, geographical location, weather) and western medicine approach

- it is also important to point out that the concept of selecting treatment based on TCM philosophy is vastly different. My own personal experience consulting TCM experts from China , which I quote him, the treatment in China (Wuhan experiencing winter that time) may not suit for countries with different climate and weather (e.g. a Southeast Asian country with hot and humid climate, with different diet practices)

- also consider that herbs, in raw form, extracted, or in different extraction medium in phytochemistry context would yield different phytocompounds, and one of the main gap here is a lack of consistency/ documentation/ quantitation/ interpretation of what is the mechanisms and bioactive compound involved

- regulatory challenges

- contribution of confounding factors such as co-morbidities, differences in western medicine used

6. Conclusion

The conclusion partly answers the objective. However, critical appraisal (as mentioned in the discussion section) would help interpret the results better and make it more relevant to the global scenario. The limitations are not only to conducting high quality studies (to which quality of studies were not actually evaluated and discussed in the discussion section), but application to the world, and consideration of knowledge gap.

7. Is the western medicine arm treatment really identical? There is no data available on what is given as western medicine and difficult to decide if they are identical, similar, or if they actually can be a confounding factor.

8. It would be good to at least describe what are the different composition of the common TCM formulations used.

But overall, I am very appreciative that this data will be made available and I look forward to the amended version. Again, I cannot emphasize enough how valuable these data are.

Reviewer #3: Reviewer’s Comments

Chinese herbal medicine in adults with mild to moderate coronavirus disease 2019(COVID-19): A systematic review and meta-analysis with MS ID PONE-D-20-38124.

Major Comments

1. Meta-analytical studies have been carried out majorly on the basis of ref 10-20 and all of them are published in Chinese journals except ref 14 only, which indicates towards the biasness of choice of content used for carrying out the study. Authors are recommended to refer the content from other sources as well to further validate the findings.

2. COVID-19 data provided in introduction section is contradictory with WHO data. Authors are suggested to cross-check the COVID-19 count provided on WHO website.

3. Conclusion of study is not in accordance with results therefore needs to be modified accordingly.

4. Manuscript mandatorily needs to be handled by language experts as there exists several ambiguities in its current form.

Minor Comments

1. Abbreviations are missing throughout the manuscript.

2. Cross-check the format of references to maintain homogeneity.

Reviewer #4: This is a very important review to publish at this time. These findings are very relevant and contribute to the essential knowledge about a globally crippling disease. The review was performed with rigorous standards and therefore the results can contribute significantly to the prevention and treatment of COVID-19. Thank you for your work.

Full review comments uploaded as attachment.

6. PLOS authors have the option to publish the peer review history of their article (what does this mean?). If published, this will include your full peer review and any attached files.

Reviewer #1: **Yes: **Muhammad G. Khodary Omar

Reviewer #2: **Yes: **Xin Yi, Lim

Reviewer #3: **Yes: **Dr. Vedpriya Arya

Reviewer #4: **Yes: **Daniela R. A. Rambaldini

---

## [Author Response · Author response to Decision Letter 0]

5 Apr 2021

Reviewer #1: kindly, find the primary review and comments in the attached Pdf file, in addition, please pay attention to the following points:

-The main claim of the paper is clear and significant, specially in such unprecedent situation.

-The analysis of data supports the claim of the paper, however; it would be better to connect this study with more previous published data and literatures in a way that reduce duplication and support the findings of this paper.

Response: in the discussion section, this review has linked this study with more previously published data and literature for analysis.

-a more detailed protocol of the statistical analysis is needed especially, most of the data used in the analysis has been retrieved from papers in Chinese language.

Response: in our review, a detailed protocol of the statistical analysis was developed. Trials on Chinese herbal medicine for mild to moderate COVID-19 were conducted in mainland China. Most of the trials were published online in Chinese. Therefore, most of the data used in the analysis has been retrieved from papers in Chinese language.

-Type of samples in treatment and control groups doesn't exclude the possibility of synergistic/ combination effect between CHM and western medicine. have you had any studies that used CHM only on separate groups as a treatment? Was there any control group that didn't receive any treatment? is there any information about hospitalization or receiving any other special care(ex. ventilator) beside the treatment?

Response: trials of Chinese herbal medicine in the treatment of mild to moderate COVID-19 were included in this review. The treatment group was treated with Chinese herbal medicine combined with conventional therapy. No trials that used CHM only on separate groups as a treatment. There was no control group that did not receive any treatment. Since the participants were diagnosed as mild to moderate COVID-19, patients did not receive ventilator treatment. The specific treatment information is listed in Table 1.

i.e: we can't conclude for sure the CHM as a separate, effective, and safe treatment for mild to moderate COVID-19. 

Response: the conclusion of this review is that Chinese herbal medicine combined with conventional therapy could be effective and safe in the treatment of adults with mild to moderate COVID-19.

Reviewer #2: Valuable data was provided in this manuscript, which are not easily assessible for international readers outside China. Hence, I have to stress that this manuscript presents precious and valuable data that will benefit the literature and improve understanding of the role of TCM in COVID-19. However, in general, I find that there is lack of clarity in definition of many things including outcome measures and treatment groups. Importantly, the discussion was superficial. There needs to be correlation between ROB, quality of study, heterogeneity and interpretation of results. Please find my suggestion as below and as specify in the attachment:

1. Strongly suggest for professional language/ scientific proof-reading to correct grammar, sentence structuring, and selection of words that are preferred to represent precise scientific writing for the entire manuscript. Kindly check for the use of oxford comma and appropriate/excessive use of connective words throughout. The authors in particular like to start sentences with the word "And". Spacing between words and symbols needs to be checked and made consistent.

Response: grammar, sentence structure, comma, and connective words have been corrected.

2. The eligibility criteria can be rewritten as inclusion and exclusion criteria clearly; or rearrange with clearer subtopics differentiation. The different levels of the subtopics in the methods needs to be clear. For example (here I am using numbers to explain an example of how the different levels needs to be clarified. It is to the authors discretion on presenting this without the numbers)

Response: the eligibility criteria have been rewritten as inclusion and exclusion criteria.

3. Specific to the methods

a. Kindly check against the PRISMA checklist- Present full electronic search strategy for at least one database (please present the combination of keywords used); Describe method of data extraction from reports (e.g., piloted forms, independently, in duplicate) and any processes for obtaining and confirming data from investigators (kindly mention if attempts were made to seek for additional data)

Response: the PubMed search strategy is listed. The method of data extraction from reports, and any processes for obtaining and confirming data from investigators were described in this review.

b. Clarify inclusion criteria- oral Chinese herbal medicine only

Response: inclusion criteria have been clarified - oral Chinese herbal medicine only.

c. Outcome measures need to be well defined e.g. what is clinical cure rate, what is effective rate of lung CT

Response: Outcome measures (e.g. clinical cure rate, lung CT) have been well defined.

4. Results

a. arrange the level of subheadings accordingly as suggested for methods

Response: the level of subheadings has been arranged accordingly as suggested for methods.

b. definition of CHM and CWM needs to be clear- the naming of the groups. Although it is mentioned that CHM group received both herbal and western medicine in methods, CHM is still abbreviated as chinese herbal medicine. The results are mostly written as 'the outcomes are better with treatment by CHM', which can be confusing to interpret, and easily misunderstood as if CHM solely (without western medicine) is beneficial. Suggest to clearly describe what each group means with distinct abbreviations for groups. Perhaps it is also because of the choice of word 'by' which when read, is interpreted this way, hence consider rewriting the results section with more precise selection of words.

Response: the naming of the groups has been rewritten.

5. Discussion

Although an interesting topic with very valuable data (I cannot emphasize this enough, this is very valuable data), the discussion is superficial and lacked depth. few suggestion of topics to discuss include

- heterogeneity of the studies and the impact on the findings.

- impact of different formulations used and how did the authors came to collectively interpreting them in the same meta-analyses (also consider that different herbs would have acted differently, and certainly herb-herb interaction should be discussed)

- risk of bias and how that affects results interpretation

- discuss on adverse events, reporting bias?

- quality of herbal intervention used

- suggest to consider consort checklist for tcm to evaluate quality of reporting which can further strengthen discussion

- how does this new information applies to the global scenario and what are the challenges of applying TCM in this scenario

- difference between TCM approach (Which is based on individualised assessment, and can be even affected by factors such as diet, body type, environment, geographical location, weather) and western medicine approach

- it is also important to point out that the concept of selecting treatment based on TCM philosophy is vastly different. My own personal experience consulting TCM experts from China , which I quote him, the treatment in China (Wuhan experiencing winter that time) may not suit for countries with different climate and weather (e.g. a Southeast Asian country with hot and humid climate, with different diet practices)

- also consider that herbs, in raw form, extracted, or in different extraction medium in phytochemistry context would yield different phytocompounds, and one of the main gap here is a lack of consistency/ documentation/ quantitation/ interpretation of what is the mechanisms and bioactive compound involved

- regulatory challenges

- contribution of confounding factors such as co-morbidities, differences in western medicine used

Response: in our review, the suggestions on the above topics have been incorporated into the discussion.

6. Conclusion

The conclusion partly answers the objective. However, critical appraisal (as mentioned in the discussion section) would help interpret the results better and make it more relevant to the global scenario. The limitations are not only to conducting high quality studies (to which quality of studies were not actually evaluated and discussed in the discussion section), but application to the world, and consideration of knowledge gap.

Response: critical appraisal has been made.

7. Is the western medicine arm treatment really identical? There is no data available on what is given as western medicine and difficult to decide if they are identical, similar, or if they actually can be a confounding factor.

Response: the western medicine arm treatment really is not identical in different trials. Specific treatment information is listed in Table 1.

8. It would be good to at least describe what are the different composition of the common TCM formulations used.

Response: the different components of TCM were described in this review.

But overall, I am very appreciative that this data will be made available and I look forward to the amended version. Again, I cannot emphasize enough how valuable these data are.

Reviewer #3: Reviewer’s Comments

Chinese herbal medicine in adults with mild to moderate coronavirus disease 2019(COVID-19): A systematic review and meta-analysis with MS ID PONE-D-20-38124.

Major Comments

1. Meta-analytical studies have been carried out majorly on the basis of ref 10-20 and all of them are published in Chinese journals except ref 14 only, which indicates towards the biasness of choice of content used for carrying out the study. Authors are recommended to refer the content from other sources as well to further validate the findings.

Response: trials of Chinese herbal medicine in the treatment of mild to moderate COVID-19 were comprehensively searched in eight electronic databases. Potentially eligible data was obtained by manually searching the reference list of previously published reviews. If possible, the conference abstracts were reviewed to find unpublished trials, and the data was obtained by contacting the author.

2. COVID-19 data provided in introduction section is contradictory with WHO data. Authors are suggested to cross-check the COVID-19 count provided on WHO website.

Response: COVID-19 data was cross-checked according to WHO website.

3. Conclusion of study is not in accordance with results therefore needs to be modified accordingly.

Response: conclusion of our study was modified in accordance with results.

4. Manuscript mandatorily needs to be handled by language experts as there exists several ambiguities in its current form.

Response: our manuscript was handled by language experts.

Minor Comments

1. Abbreviations are missing throughout the manuscript.

Response: abbreviations full names were listed in the manuscript.

2. Cross-check the format of references to maintain homogeneity.

Response: the format of references was cross-checked.

Reviewer #4: This is a very important review to publish at this time. These findings are very relevant and contribute to the essential knowledge about a globally crippling disease. The review was performed with rigorous standards and therefore the results can contribute significantly to the prevention and treatment of COVID-19. Thank you for your work.

---

## [Decision Letter · Decision Letter 1]

24 May 2021

PONE-D-20-38124R1

Chinese herbal medicine in adults with mild to moderate COVID-19: A systematic review and meta-analysis

PLOS ONE

Dear Dr. Shi,

Thank you for submitting your manuscript to PLOS ONE. After careful consideration, we feel that it has merit but does not fully meet PLOS ONE’s publication criteria as it currently stands. Therefore, we invite you to submit a revised version of the manuscript that addresses the points raised during the review process.

We look forward to receiving your revised manuscript.

Kind regards,

Ahmed Negida, MD

Academic Editor

PLOS ONE

Journal Requirements:

Reviewers' comments:

Reviewer's Responses to Questions

**Comments to the Author**

1. If the authors have adequately addressed your comments raised in a previous round of review and you feel that this manuscript is now acceptable for publication, you may indicate that here to bypass the “Comments to the Author” section, enter your conflict of interest statement in the “Confidential to Editor” section, and submit your "Accept" recommendation.

Reviewer #2: (No Response)

Reviewer #4: All comments have been addressed

2. Is the manuscript technically sound, and do the data support the conclusions?

Reviewer #2: Yes

Reviewer #4: Yes

3. Has the statistical analysis been performed appropriately and rigorously? 

Reviewer #2: Yes

Reviewer #4: Yes

4. Have the authors made all data underlying the findings in their manuscript fully available?

Reviewer #2: Yes

Reviewer #4: Yes

5. Is the manuscript presented in an intelligible fashion and written in standard English?

Reviewer #2: No

Reviewer #4: Yes

6. Review Comments to the Author

Reviewer #2: Thank you for revising your very valuable manuscript. It was really interesting and my honour to have read such valuable data. It is really precious. However, major revisions especially in data interpretation, critical analysis and discussion writing, as well as definition of outcomes still needs to be amended.

1) Introduction - need to better justify the need for this systematic review (i.e research gap leading to the objective). It is too brief. In fact the introduction in abstract better justified the purpose of the review than the introduction in the manuscript. Suggest to amend. Also, very importantly, there is an overstatement on the findings when citing two studies on benefits of CHM in the introduction, please review again as commented in the manuscript).

2) Methods and results: Please define in depth for outcomes : lung CT- what is the definition of improved/good lung CT etc, please elaborate. and how does that relate to outcome assessment. The same applies to definition of clinical cure rate and viral nucleic acid testing. The definition of effective, improved, ineffective also needs to be clarified

3) Results- table 2 is truncated and I could not view the entire table. The resolution of the forest plots were low and many are not readable. I suggest to rename the axis to favour therapy alone and favour CHM + conventional therapy instead of favour control/favour experimental to make it easier to read at first glance.

4) The benefits on inflammatory markers- this is very important to be clarified. Is higher levels better or lower levels better? This would depend on the course of the disease. For example, at initial stages, we probably need higher levels of immunity to produce antiviral effects. However, if inflammation is overt and prolonged at later stages, this may push the patient into a cytokine storm and hence detrimental i.e. timing is important. This component is not discussed and should be discussed, considering that inflammatory markers is one of the outcomes assessed and duration of administration varies .

5) I would appreciate more discussion on CHM+ conventional therapy discussion in view that this plays a major role in your findings too

6) English has improved but still needs to be improved further. In particular in selection of accurate and politically appropriate words, and an entire highlighted paragraph in discussion. Minor formatting changes of the tables are also recommended.

7) Data availability statements should be clarify- its contradicting with what is already provided.

Reviewer #4: Authors seem to have addressed all recommendations and points including clarification of methods, improvements in the statistical analysis and tests used, updates to the results, revisions to the discussion, and overall edits to grammar and sentence structure. Addition of Tables 1 & 2 are very helpful.

7. PLOS authors have the option to publish the peer review history of their article (what does this mean?). If published, this will include your full peer review and any attached files.

Reviewer #2: No

Reviewer #4: **Yes: **Daniela R. A. Rambaldini

---

## [Author Response · Author response to Decision Letter 1]

8 Jul 2021

Reviewer #2： Thank you for revising your very valuable manuscript. It was really interesting and my honour to have read such valuable data. It is really precious. However, major revisions especially in data interpretation, critical analysis and discussion writing, as well as definition of outcomes still needs to be amended.

1) Introduction - need to better justify the need for this systematic review (i.e research gap leading to the objective). It is too brief. In fact the introduction in abstract better justified the purpose of the review than the introduction in the manuscript. Suggest to amend. Also, very importantly, there is an overstatement on the findings when citing two studies on benefits of CHM in the introduction, please review again as commented in the manuscript).

Response: the need for this systematic review has been further illustrated. In the introduction, the authors have stated the specific benefits of CHM combined with conventional therapy in the treatment of mild to moderate COVID-19.

2) Methods and results: Please define in depth for outcomes: lung CT- what is the definition of improved/good lung CT etc, please elaborate. and how does that relate to outcome assessment. The same applies to definition of clinical cure rate and viral nucleic acid testing. The definition of effective, improved, ineffective also needs to be clarified

Response: the authors have defined in depth for outcomes of lung CT, clinical cure rate, and viral nucleic acid testing.

3) Results- table 2 is truncated and I could not view the entire table. The resolution of the forest plots were low and many are not readable. I suggest to rename the axis to favour therapy alone and favour CHM + conventional therapy instead of favour control/favour experimental to make it easier to read at first glance.

Response: the authors have modified table 2 and the resolution of the forest plots. The authors have renamed the axis to favour conventional therapy and favour combination therapy instead of favour control/favour experimental to make it easier to read at first glance.

4) The benefits on inflammatory markers- this is very important to be clarified. Is higher levels better or lower levels better? This would depend on the course of the disease. For example, at initial stages, we probably need higher levels of immunity to produce antiviral effects. However, if inflammation is overt and prolonged at later stages, this may push the patient into a cytokine storm and hence detrimental i.e. timing is important. This component is not discussed and should be discussed, considering that inflammatory markers is one of the outcomes assessed and duration of administration varies.

Response: the benefits on inflammatory markers have been clarified. Subgroup analyses of outcomes of inflammatory markers were carried out according to treatment duration.

5) I would appreciate more discussion on CHM+ conventional therapy discussion in view that this plays a major role in your findings too

Response: this review has discussed combination therapy of CHM and conventional therapy.

6) English has improved but still needs to be improved further. In particular in selection of accurate and politically appropriate words, and an entire highlighted paragraph in discussion. Minor formatting changes of the tables are also recommended.

Response: the authors have received linguistic assistance provided by AJE (https://secure.aje.com/cn/researcher/) during the preparation of this manuscript. The format of the tables has been modified.

7) Data availability statements should be clarify- its contradicting with what is already provided.

Response: the authors have clarified the data availability statements.

---

## [Editor Report · Decision Letter 2]

9 Aug 2021

Add-on effect of Chinese herbal medicine in the treatment of mild to moderate COVID-19: A systematic review and meta-analysis

PONE-D-20-38124R2

Dear Dr. Shi,

We’re pleased to inform you that your manuscript has been judged scientifically suitable for publication and will be formally accepted for publication once it meets all outstanding technical requirements.

Kind regards,

Ahmed Negida, MD

Academic Editor

PLOS ONE
---

## [Editor Report · Acceptance letter]

13 Aug 2021

PONE-D-20-38124R2 

Add-on effect of Chinese herbal medicine in the treatment of mild to moderate COVID-19: A systematic review and meta-analysis 

Dear Dr. Shi:

I'm pleased to inform you that your manuscript has been deemed suitable for publication in PLOS ONE. Congratulations! Your manuscript is now with our production department. 

Kind regards, 

on behalf of

Dr. Ahmed Negida 

Academic Editor

PLOS ONE